# Modified Nanoparticles as Potential Agents in Bone Diseases: Cancer and Implant-Related Complications

**DOI:** 10.3390/nano10040658

**Published:** 2020-04-01

**Authors:** Karol P. Steckiewicz, Iwona Inkielewicz-Stepniak

**Affiliations:** Chair and Department of Medical Chemistry, Faculty of Medicine, Medical University of Gdansk, ul. Dębinki 1, 80-211 Gdansk, Poland; karol.steckiewicz@gumed.edu.pl

**Keywords:** nanotechnology, nanoparticles, osteosarcoma, antimicrobial properties, nanotoxicology, biocompatibility, bone diseases, implant-related infections

## Abstract

Materials sized 1–100 nm are the nanotechnology’s field of interest. Because of the unique properties such as the ability to penetrate biological barriers and a high surface to volume ratio, nanoparticles (NPs) are a powerful tool to be used in medicine and industry. This review discusses the role of nanotechnology in bone-related issues: osteosarcoma (bone cancer), the biocompatibility of the implants and implant-related infections. In cancer therapy, NPs can be used as (I) cytotoxic agents, (II) drug delivery platforms and (III) in thermotherapy. In implant-related issues, NPs can be used as (I) antimicrobial agents and (II) adjuvants to increase the biocompatibility of implant surface. Properties of NPs depend on (I) the type of NPs, (II) their size, (III) shape, (IV) concentration, (V) incubation time, (VI) functionalization and (VII) capping agent type.

## 1. Introduction

Miniaturisation affects every aspect of human life; medicine and science are no exceptions. Nanotechnology is interested in particles within the 1–100 nm size range [1]. For a better understanding of the size range in Figure 1 we compare nano size to other objects. Although it was Richard Zsigmondy who used the term ‘nanometre’ as early as in 1925, Richard Feynman is the indisputable father of nanotechnology [1]. In 1959 he gave a lecture entitled ‘There’s Plenty Room at the Bottom’ and suggested that manipulation on the atomic level would soon be possible. However, the term ’nanotechnology’ was unknown until the seventies. Norio Taniguchi is thought to be the first to use it [1]. Almost a century after its beginning, nanotechnology is a rapidly developing branch of science. In 2015, nontechnology industry employed 7 million people and was worth $1 billion [2,3]. Nanoscale, because of quantum effects, causes nanoparticles (NPs) to have different properties than macromolecules. NPs have a large surface to volume ratio, and the ability to penetrate cellular membranes and structural barriers, which greatly expand its potential applications [4]. NPs are used in biology, genetic engineering, medicine, biotechnology and industry (Figure 2) [4,5,6]. Moreover, the ability to modify NPs (size, shape, surface functionalisation, capping agent) increases their potential [4]. 

In this review, we discuss the role of nanotechnology in the novel treatment of bone diseases. The human body has over 206 bones, which serve a variety of functions: locomotion, protection of internal organs, ion homeostasis and blood cells production [7,8,9]. Unfortunately, every bone can suffer from diseases and be the cause of health-related issues. Implantation-related issues and bone neoplasm have been taken in concerns. 

## 2. Cancer

Cancer is one of the leading causes of death in developed countries. Just in 2018 more than 18 million new cancer cases were diagnosed worldwide. Furthermore, cancer was the cause of death for more than 9.5 million people [10]. Between 1987 and 2005 cancer treatment costs have doubled and reached almost $50 billion in the United States alone [11]. Although primary bone cancers are relatively rare (7% of new neoplasm cases in adolescents), bone metastases happen often and by causing excruciating pain severely decrease the patients’ quality of life in end-stage disease [12,13]. 

### 2.1. Osteosarcoma

Although osteosarcoma (OS) is a primary mesenchymal bone neoplasm characteristic for the paediatric population, it can occur at any age [14]. Unfortunately, in the elderly the survival rate is roughly 2–8 times lower than in adolescents [15]. Even though OS is objectively rare (3.5–4 cases/million population/year), it is the third most common cancer in children [15]. OS is typically located in expeditiously growing long bones (femur, tibia, humerus) [15]. Less typical locations such as skull, chest or pelvis are unfavourable prognostic factors [15]. As OS quickly gives distant metastases, the disease is often already advanced at diagnosis. Lungs are the most common location of OS metastases [14]. Apart from therapeutic radiation (as treatment of previous cancer), no other risk factors of OS are known and neither are prevention methods [15]. Interestingly, OS occurs more frequently in some genetic diseases (Li-Fraumeni Syndrome, Retinoblastoma, Werner Syndrome, Bloom Syndrome, etc.,) [15]. OS treatment consists of chemo and/or radiotherapy followed by surgery. In chemotherapy methotrexate (MTX), doxorubicin (DOX), cisplatin (CDDP) and ifosfamide (IFO) are used [14]. Unfortunately up to 40–50% of OS tumours are chemo-resistant [16]. Several mechanisms are known to cause multidrug resistance (MDR) in cancer cells, i.e., enhanced detoxification, efflux pumps, decreased drug uptake and up-regulation of DNA repair mechanism [17]. Therefore, the outcome of the treatment is often poor with a 5-year survival rate of 55% (5.7–86.8% as it is localisation-dependent) [15]. Tumour recurrence due to incomplete resection and lung metastases are noted as the leading reasons for treatment failure [18]. It is worth emphasising that the current treatment protocols severely impair patients’ quality of life. Therefore, novel approaches to OS are searched. 

### 2.2. Nanoparticles Cytotoxicity to Osteosarcoma Cells 

In literature, several papers suggest the anticancer activity of NPs. Rahim et al. have shown that 24.3 nm gold nanoparticles (AuNPs) capped with advance glycation products can decrease cell viability and trigger apoptosis in Saos-2 (osteosarcoma) cells [19]. Interestingly, other studies suggested that the anticancer activity of AuNPs is shape dependent. 143b and MG63 osteosarcoma cells were sensitive to AuNPs rods and stars but not to AuNPs spheres [20]. AuNPs are not the only NPs with anticancer activity. AgNPs can decrease the viability of the MG63 (osteosarcoma) cells [21]. The question is whether the observed effect is nano-size-related or due to the presence of silver. It was shown that 15–34 nm AgNPs are more cytotoxic than AgNO_3_ to the A-431 (osteosarcoma) cells [22]. Likewise, Kovacs et al. have shown that 5 nm and 35 nm citrate-AgNPs influenced the viability of two osteosarcoma cell lines U2OS and Saos-2 [23]. They have shown that cytotoxicity is size-dependent: the smaller the AgNPs were, the stronger their cytotoxic abilities. Moreover, AgNPs also inhibited cell proliferation and were more effective than cisplatin in the same concentration. AgNPs act by triggering mitochondrial stress and eventually, apoptosis [23]. Another metal with anticancer activity in nano-form is copper. Copper nanoparticles (CuNPs) embedded in alginial hydrogel in a concentration of >0.5% wt. decreased viability of the Saos-2 cells [24]. Unfortunately, there is no literature data on the mechanism of CuNPs cytotoxicity in osteosarcoma cells. We were unable to find any data about the impact of iron or aluminium nanoparticles on osteosarcoma cells either. 

Additionally, metal oxide nanoparticles can have anticancer activity. It has been shown that 3.8 nm titanium oxide nanoparticles (TiO_2_NPs) in the concentration of >0.5 μg/mL were cytotoxic against the U2OS cells in a time- and concentration-dependent manner. TiO_2_NPs induced excessive ROS production and depletion of glutathione (GSH), triggering oxidative stress [25]. In another study, cytotoxicity of TiO_2_NPs was also confirmed. Di Virgil et al. examined the anticancer activity of 15 nm TiO_2_NPs and 50 nm aluminium oxide nanoparticles (Al_2_O_3_NPs) [26]. Both NPs types were cytotoxic against the UMR-106 cells in the concentration of >50 μg/mL (MTT assay) [26]. Among others, pH is one of the factors influencing cancer cells response to NPs. It was reported that 3–4 nm dextran-coated cerium oxide nanoparticles (CeO_2_NPs) were more effective against osteosarcoma cells in acid pH (pH = 6) than other pH levels (pH = 7, pH = 9). Interestingly, in the same condition, cytotoxicity of CeO_2_NPs to non-cancerous bone cells was minimal. The study suggested increased ROS production as a mechanism of CeO_2_NPs cytotoxicity [27]. This observation was confirmed in another study which proved that zinc oxide nanoparticles (ZnONPs) could be harmful to MG63 as they triggered ROS production [28].

Not only metal nanoparticles can be used against osteosarcoma. Kimura et al. showed that fucoidan nanoparticles (100 nm) in the concentration of 1–8 mg/mL decreased the viability of the 143B cells by triggering apoptosis [29]. In the study, fucoidan NPs had higher anticancer activity than macro-size fucoidan in CH3 mice in vivo osteosarcoma model. Interestingly, as in the in vitro model, fucoidan NPs triggered apoptosis in osteosarcoma in vivo as well. Moreover, fucoidan NPs did not affect the bodyweight of the animals, therefore they should not have severe side effects [29]. Also, hydroxyapatite nanoparticles (HA-NPs) were shown to have beneficial properties. HA-NPs are especially interesting because of the similarity of their composition and crystal structure to the microarchitecture of a bone [30]. Interestingly, it was shown that HA-NPs can induce apoptosis in the MG63 cells and promote the viability of healthy osteoblasts [30]. Beside selective cytotoxicity only to cancer cells, HA-NPs also caused ultrastructure changes. Swollen mitochondria, ribosome detachment from rough endoplasmic reticulum, and changes in nuclear morphology were observed [30]. 

For better understanding of NPs biological activity, it is essential to know whether NPs are being internalised or not. It has been shown that different nanoparticles can be uptaken and accumulated by osteosarcoma cells. Azarami et al. have proven that the uptake of 112–303 nm gelatine nanoparticles by the 143B cells is size-dependent. The larger the nanoparticles were, the less efficiently they were internalised [31]. Similarly, it was shown that 100 nm PGLA NPs can be internalised by the MG63 cells. PLGA NPs were internalised by endocytosis and accumulated in the cytoplasmic region [32]. 

To sum up, different nanoparticles (metal, metal oxide, HA) can have anticancer activity, typically mediated by increased ROS production. Modification of NPs such as size, shape, type of NPs and/or capping agent can affect their anticancer activity of NPs. The summary of NPs anti-osteosarcoma activity is presented in Table 1. 

### 2.3. Nanoparticles Cytotoxicity to Other Bone Cancer Types 

Chondrosarcoma, Ewing’s sarcoma and fibrosarcoma are other types of cancers, however they are far less common than OS. Unfortunately, data about NPs cytotoxicity against them is limited. Sha et al. examined the effect of 3.8 nm TiO_2_NPs on the SW1353 chondrosarcoma cells [25]. They observed time- and concentration-dependent cytotoxicity of TiO_2_NPs. Interestingly, the chondrosarcoma cells in the study were more susceptible to NPs than the osteosarcoma cells (U2OS). Authors suggested the induction of oxidative stress as TiO_2_NPs cytotoxicity mechanism [25]. NPs were also used as a strategy to treat Ewing’s sarcoma. Elhamess et al. used genetically modified NIH/3T3 cells as Ewing sarcoma model in which they have shown that oligonucleotides-chitosan nanospheres may be an efficient gene delivery platform [33]. A summary of NP’s effect on fibrosarcoma in vitro model is presented in Table 2.

### 2.4. Nanoparticles as Drug Delivery Platforms 

NPs as drug delivery platforms have a lot of advantages: improved efficiency, reduced toxicity, smaller cost of therapy, potential effectiveness in MDR cancers [31]. It has been shown that NPs can accumulate in the cancer microenvironment because of the improper structure and function of endothelial cells in the tumour vasculature (wider junctions, fenestration, incomplete basal membrane) which makes it easier to penetrate [41]. This observation was called enhanced permeability and retention effect, and it is probably the basis of NPs anticancer effect [42]. The summary of all NPs as drug delivery platforms is presented in Table 3. 

Dhule et al. have shown that liposomal NPs can be used as curcumin drug delivery platforms [43]. Curcumin is not yet being used in clinical practice, however, its anticancer effect is well established and cancer cells are more susceptible to curcumin than non-transformed ones [43]. Moreover, liposomal NPs with curcumin trigger apoptotic death of KHOS (osteosarcoma) cells, whereas curcumin alone induces autophagy [43]. It proves that wisely used drug delivery platforms can change compound properties to more favourable ones. Also, Shu-Fen et al. have shown the effectiveness of curcumin. Their 250 nm curcumin-loaded PGLA NPs significantly decreased the viability of U2OS cells [16]. In that study, curcumin in NPs induced apoptotic osteosarcoma cell death by triggering mitochondria-dependent apoptosis [16]. Curcumin was not the only drug to be conjugated with NPs. Ni et al. designed 150 nm spherical, salinomycin-loaded PEG nanoparticles with aptamer to target osteosarcoma stem cells [18]. Salinomycin is an old chemotherapeutic drug with high anticancer stem cells activity [18]. Unfortunately, its potential is greatly reduced by its water insolubility [18]. Salinomycin loaded PEG NPs were effective against Saos-2, U2OS and MG63 cells and even more effective against cancer stem cells (CD133 positive) [18]. Moreover, in Saos-2 population, cancer stem cells were greatly reduced by adding an aptamer to salinomycin PEG-NPs treatment [18]. Those findings were also confirmed in in vivo model. In Balb/c mice with an osteosarcoma tumour (from Saos-2 cells) treated with NPs had their tumour weight, number of mammospheres formed and amount of cancer stem cells reduced compared to control [18]. NPs loaded with two different cytostatic were also studied. Wang et al. created complex NPs. They encapsulated paclitaxel (PTX) and etoposide (ETP) in 100 nm PEG-ylated PLGA nanoparticles (PTX-ETP/PLGANPs). Plain NPs (without PTX or ETP) were not cytotoxic, which proves the safety of application [32]. The nanoparticles were more effective against MG63 and Saos-2 cancer cells than PTX or ETP in combination, which demonstrated that nano form significantly changes the properties of NPs. In a more detailed analysis it was demonstrated that PTX-ETP/PLGANPs are more effective in inducing MG63 cell apoptosis than drugs without carrier [32]. Some scientists went further and combined chemotherapy with gene therapy to overcome drug resistance. Sun et al. prepared 200 nm dextran-g-PEI NPs (DEX-PEI NPs) to be an adriamycin (ADM) and plasmid transporter. They have shown that DEX-PEI-ADM NPs were more cytotoxic against the MG63 and Saos-2 osteosarcoma than ADM or DEX-PEI NPs [41]. Next, they examined properties of DEX-PEI-ADM NPs as a plasmid carrier. They tried to express a green fluorescent protein (GFP) in the MG63 and Saos-2 cells. GFP was chosen, as it is easy to determine whether the transfection was effective or not. DEX-ADM-PEI NPs with GFP pcDNA turned out to be an effective transfection reagent [41]. However, NPs were less effective than Lipofectamine 2000, a typically used transfection reagent (transfection effectiveness for NPs were 18.6% and 15.3% for MG63 and Saos-2 cells respectively, whereas for Lipofectamine 2000 it was 26.6% and 21.8%). Also, Susa et al. established DEX-containing NPs. They created 112.4 nm stearylamine-dextran nanoparticles loaded with DOX (STE-DEX-DOXNPs) [17]. They examined STE-DEX-DOXNPs on the U2OS, KHOS and MDR osteosarcoma cell lines. Interestingly, after treatment with NPs DOX were more accumulated in the drug-resistant cancer cell lines than in the regular KHOS or U2OS cells. Moreover DOX in a free form accumulated in osteosarcoma cells cytoplasm, whereas STE-DEX-DOXNPs were trafficked to the nucleus of the cells [17]. Also, STE-DEX-DOXNPs have an antiproliferative effect and caused apoptosis of OS cells. This effect was more prominent than in cells treated only with DOX. 

### 2.5. Magnetic Nanoparticles 

Hyperthermia defined as the treatment of cancer with heat is a well-established practice. It is proven that cancer cells are more susceptible to heat and in the temperature > 43 °C they undergo necrosis [45]. The main problem of this approach is the impossibility to provide heat only to the tumour and avoid healthy tissues. The use of magnetic nanoparticles that can be directed to the tumour and then heated could enable overcoming that issue [46]. The most clinically promising method of NPs heating is capacitive heating using a radiofrequency electric field [47]. Makridis et al. have suggested 26 nm Mn-Fe_2_O_4_ NPs in cancer treatment. They have proven that Mn-Fe_2_O_4_ NPs were internalised by Saos-2 cells in energy-dependent endocytosis, also the cancer cells were more susceptible to heating than the non-transformed ones. The magnetic field used to heat nanoparticles was not harmful to the cells [46]. Hyperthermic treatment’s effectiveness was also proven in vivo. Matsuoka et al. created magnetic cationic liposomes (MCL) based on supramagnetic iron oxide nanoparticles [48]. They injected MCL directly into an osteosarcoma tumour in a female Syrian hamster. Next, tumour was heated to above 42 °C. They observed >15 days regression in all tested animals (75% animals had complete regression). Moreover, tumour mass in treated animals was 0.1% (1/1000) of tumour mass in control subjects [48]. A similar observation was made by Shido et al. [47]. They also used MCL and heated the tumour to above 43 °C. They used C2/He mice model. They were able to achieve suppression of tumour growth in all treated animals, with complete regression in 43% of treated animals, whereas in control animals tumour volume was increasing over time [47]. The group which underwent treatment also presented less metastases in comparison to control animals (mean number of lung metastasis 56.8 versus 17.6) [47]. Interestingly, However, magnetic NPs are used not only in hyperthermic treatment. Xeu-Song et al. created poly-lactic acid arsenic trioxide nanoparticles (ATONPs) [49]. Arsenic trioxide is a compound used in the treatment of acute promyelocytic leukaemia. In the study, they created 60–70 nm magnetic ATONPs and examined their anticancer abilities. They have shown that with the usage of a magnetic field, ATONPs can be directed to a specific place. They observed 40% higher concentrations of ATONPs in the kidneys of a New Zealand white rabbit if magnetic field was used [49]. Moreover, the ATONPs were effective against osteosarcoma in in vivo model (BALB/c nude mice). Also, Kubo et al. used magnetic liposomes in drug delivery. They created 146 nm magnetic liposomes incorporated with adriamycin (MLA) [50]. In vivo assessment (Syrian hamster) have shown that only MLA under magnetic force were able to suppress tumour growth [50]. To summarise, magnetic NPs in OS treatment can be used two-fold: as hyperthermic agents or delivery platforms. In both approaches, NPs are effective both in vitro and in vivo.

## 3. Nanoparticles in Orthopaedic Implants 

Because medical advancement societies are aging, it brings forward new health issues such as osteoarthritis for which conservative treatment is often not sufficient and joint replacement surgery is needed. Unfortunately, epidemiological data are terrifying. Ten percent of >15 years old Canadians suffer from osteoarthritis. Almost half of the population at the age of 65 or older has osteoarthritis of at least one joint [51]. Pain and movement impairment are the most prominent symptoms, severely decreasing patient’s quality of life. It is the obligation of the scientific community to address main issues regarding joint replacement surgery: implant-related infections and poor biocompatibility. NPs may both increase biocompatibility of the implants and be the prophylaxis of the implant-related infections [52,53]. This knowledge can be also used in other orthopaedic implantable devices such as artificial ligaments or tendons, bone nails or screws. 

### 3.1. Implant-Related Infections

In total, half of all nosocomial infections are related to implantable devices [54]. Although not very common among orthopaedic patients (2–5%), this complication is costly ($1.86 billion annually only in the United States) [54,55], especially if we consider that 500,000 people have a hip or knee replacement in the United States alone [56]. Infections related to orthopaedic devices are often a cause of their failure, leading to another surgery [54]. They can also be a facilitator for other serious complications and even death, as they increase the risk of cardiac arrest, pulmonary embolism, myocardial infection and acute renal failure [57]. Age, obesity and comorbidities are the main risk factors of serious complications [57]. Moreover, the diagnosis of implant-related infection is complicated and the symptoms may occur many months after surgery [56]. Treatment typically consists of broad-spectrum systemic antibiotic treatment and removal of the infected implant [54]. Typically infection is caused by aerobic Gram-positive bacteria such as *Staphylococcus aureus* (34%), *Streptococcus epidermidis* (32%) and other coagulases negative streptococci (13%) [54]. However, Gram-negative pathogens (*Pseudomonas* spp., *Enterococcus* spp., *Escherichia* spp.) and fungi (*Candida* spp., *Aspergillus* spp.) also may cause the infection [54,56,58]. Planktonic bacteria are far less dangerous than the ones forming a biofilm. A biofilm is defined as a complex structure made of bacterial cells and extracellular matrix, which allows cells to exchange virulence factors via plasmids [59]. Importantly pathogens in the biofilm are more resistant to treatment than planktonic form, typically 100–1000 times [60]. Moreover, neither antibiotics nor immune cells can penetrate a biofilm easily, which makes the treatment challenging. It can occur on any surface that bacterial cells can adhere to, orthopaedic devices included [59]. 

### 3.2. Biocompatibility 

Bone is a metabolically active tissue with a great remodelling potential [7]. Therefore, it is important that the implant or any other device is incorporated into the surrounding tissue. Titanium is one of the most popular implant materials, so naturally, the modification of titanium surface to increase its biocompatibility is the most popular choice. Ren et al. proposed titanium-AgNPs-titanium nanostructure [61]. Such nanostructures have good antimicrobial properties; moreover, the MC3T3-E1 cells (mouse preosteoblast) attached and proliferated on the nanostructure easily. Preosteoblast had proper morphology and appropriate amount of alkaline phosphatase (ALP) activity which is a marker of osteogenesis [61]. Unfortunately, high concentration of nanostructure and prolonging incubation impacted the cellular proliferation and morphology [61]. A strategy to enhance antimicrobial properties of the implants was also used by Xiang et al. They used poly(lactic-*co*-glycolic acid)/ZnO nanorods/Ag nanoparticles hybrid coating on Ti implants (PLGA-ZnO-AgNPs) [62]. Their coating had antimicrobial activity against both Gram-positive and Gram-negative bacteria (*Staphylococcus aureus*, *Escherichia coli*). A biocompatibility assessment showed that PLGA-ZnO-AgNPs were less cytotoxic than ZnO or ZnO-Ti against MC3T3-E1 cells [62]. They also used ALP as a marker of osteogenesis and an increased level of that enzyme in cells on the PLGA-ZnO-AgNPs surface was noted [62]. As further confirmation they observed the proper formation of cytoskeleton within the MC3T3-E1 cells [62]. Also, Neupane et al. modified titanium nanotubes with AuNPs (TiO_2_-AuNPs). They compared TiO_2_-AuNPs to polished Ti (Ti_p_) and TiO_2_ nanotubes (TiO_2_NPs). In comparison to other materials, the MC3T3-E1 cells on the surface of TiO_2_-AuNPs had more visible nuclei and more filopodia, and therefore higher osteoblast activity [53]. It was further confirmed by MTT assay that the MC3T3-E1 cells were more viable when treated with TiO_2_-AuNPs than Ti_p_ or TiO_2_NPs [53]. In another paper, titanite nanotubes were modified with AgNPs [63]. The created material was expected to have antibacterial properties against *Escherichia coli* [63]. In comparison to the titanium control the proposed coating did not affect MC3T3-E1 proliferation, moreover, it promoted cells adhesion and migration [63]. Hydroxyapatite (HA) may promote the proliferation of healthy bone cells [30], and thus several attempts were made to functionalise the implants with HA. Fomin et al. functionalised titanium surface with hydroxyapatite nanoparticles (HA-NPs). They have found that this modification improved fibroblast fixation to the surface [64]. Salaie et al. modified medical titanium alloy with AgNPs and HA-NPs [65]. Unfortunately, their coating was slightly toxic (cell viability was decreased by around 30%), however, in a morphological analysis cells showed no signs of distress and filopodia formed well [65]. Those findings have proven that the modifications of implants surface with NPs may increase their biocompatibility and act as an antimicrobial agent. They had shown so much promise that some of the modifications were even patented [66].

### 3.3. Nanoparticles in Bone Regenerative Strategies

Because of their unique properties, some NPs may promote osteogenesis. Wei et al. reported that AgNPs promoted osteogenesis by inducing autophagy [67]. AgNPs assessed in human mesenchymal stem cells model in a non-toxic concentration were internalised, promoted osteogenesis (increased mineralisation and alkaline phosphatases activity) and matrix protein synthesis [67]. 53-nm AuNPs modified with advanced-platelet-rich-plasma were non-cytotoxic and promoted osteogenesis (by increasing alkaline phosphates activity and calcium content) [68]. Patel et al. have created hydroxyapatite NPs (HA-NPs) and examined their effect on bone marrow-delivered mesenchymal stem cells (BMSCs) [69]. They have proven that HA-NPs were non-toxic to BMSCs and promoted osteogenesis (increased level of calcium and gene expression of osteoblast markers) [69]. NPs made of hydroxyapatite and gold (HA-Au-NPs) had particularly beneficial properties. Liang et al. have shown that HA-Au-NPs were internalised by endocytic pathway and promoted osteogenesis, [70]. Increased alkaline phosphatase activity and expression of osteogenic genes were reported. Authors suggested that the observed effect was Wnt/ß-catenin pathway-dependent [70]. In another study HA-NPs were enriched in Li^+^ ions [71]. The created biomaterial promoted osteogenesis and mitochondrial dynamic and inhibited apoptosis in adipose tissue-derived mesenchymal stem cells model [71]. Also, calcium polyphosphate NPs (polyP-NPs) can stimulate osteogenesis. Hatt et al. have proven that polyP-NPs can be a source of phosphate for matrix mineralisation and increased osteogenesis marker levels [72]. Graphene oxide may be an interesting biomaterial too. Several studies have shown that it has the abilities to promote osteogenesis and it can also be effective against *Staphylococcus aureus* [73,74]. Pro-osteogenic properties of NPs have also been proven in in vivo model. Wang et al. have reported that aptamer-functionalised NPs (AP-NPs) may increase the osteogenesis markers level (osteopontin, osteocalcin, alkaline phosphatase) and improve the femur bone regeneration [75]. Moreover, AP-NPs were non-toxic in in vitro BMSCs model. Also, sinopic acid-loaded chitosan NPs (SA-CH-NPs) promoted osteogenesis in vivo (observed as better regeneration of cervical bone) [76]. Those results corresponded with in vitro assessment, where NPs were non-toxic and promoted osteoblast formations from BMSCs through activation of the TGF-ß1/BMP/Smads/Runx2 pathway [76]. Study designed by Kuang et al. is especially interesting because they created an injectable material containing nanocomposite hydrogel and CaPNPs [77]. The injectable material was potentially convenient to use and its effectiveness in promoting osteogenesis was proven both in vitro and in vivo [77]. To summarise, both organic and inorganic NPs can promote osteogenesis and be non-toxic to mammalian cells. These abilities may be used in regenerative medicine. 

### 3.4. Antimicrobial Properties of Nanoparticles 

AgNPs are the ones with the best-described antimicrobial activity. Baker et al. have shown that 75 nm AgNPs can be effective against *Escherichia coli* [52]. However, the antibacterial properties of silver are size-dependent. The study has proven that the smaller AgNPs (7 nm) were more effective than the bigger ones (29 nm, 89 nm). That observation was made by a comparison of minimal inhibitory concentration (MIC) of two bacterial strains *E. coli* and *Staphylococcus aureus* [78]. Another paper showed that AgNPs are more effective against Gram-negative bacteria than against Gram-positive ones [79]. Moreover, AgNPs can also act against drug-resistant bacteria (ampicillin-resistant *Escherichia coli* and multi-drug resistant *Salmonella typhi*) [79]. In other studies, AgNPs inhibited the growth of *Bacillus subtilis, Klebsiella mobilis, Vibrio cholera, Pseudomonas aeruginosa, Shigella flexneri, Mycobacterium smegmatis* and *Mycobacterium tuberculosis* [5,80,81]. It was only the size that influenced the antimicrobial properties of AgNPs. Niska et al. examined the role of the capping agent on antimicrobial properties of AgNPs [82]. They examined uncapped AgNPs, AgNPs capped with lipolic acid (LA), tannic acid (TA) or PEG. UC-AgNPs and LA-AgNPs had the strongest antimicrobial activity, whereas TA-AgNPs the smallest. Their AgNPs had also an antibiofilm activity. In their study, Gram-positive strains were more susceptible to AgNPs which is contrary to the findings of Shrivastava et al. [79,82]. Several mechanisms of AgNPs antimicrobial properties are suggested (Figure 3); the inhibition of transduction of signalling pathways, lytic effect on the cellular membrane, increased ROS production, inhibition of enzymes, inactivation of nucleic acids are worth mentioning [79,83,84,85]. AgNPs can also have an antifungal activity [84]. It was reported that 25-nm AgNPs inhibited the growth of four strains of *Candida* spp., AgNPs were used in concentrations non-cytotoxic for mammalian cells [86]. Moreover, stabilisation with surfactants or polymers improved the antifungal activity of AgNPs [86]. This observation was in accordance with other studies, and also found that 3-nm AgNPs are effective against *Trichophyton mentagrophytes* [87]. AgNPs were more effective than commonly used medication: amphotericin b and fluconazole [87]. AgNPs possibly have antiviral and antiprotozoal activity as well, however viruses and protozoa almost never cause bone infections [88,89]. 

Also, AuNPs can be an interesting antimicrobial agent. Cui et al. reported the antibacterial activity of AuNPs against *E. coli* [90]. AuNPs inhibited the growth of both planktonic form and biofilm [90]. AuNPs impacted the expression of 359 genes, decreased ATP concentration within the bacterial cells and triggered ROS production [90]. Also, Gram-positive bacteria may be susceptible to AuNPs. The same paper has shown that 11–22 nm AuNPs can be an antifungal agent against *Candida* spp. and *Aspergillus* spp. [91]. Transmission electron microscopy (TEM) has shown that AuNPs attached themselves to bacterial cells and caused improper respiration and permeability [91]. Other papers also supported that AuNPs can be an antibacterial agent [92]. Unfortunately, some studies did not prove the antibacterial properties of AuNPs [93,94]. NPs types such as copper, zinc oxide, titanium oxide and others can also have antimicrobial properties [95,96,97,98,99]. A more detailed description of those NPs antimicrobial properties is presented in Table 4. NPs may be potentially used as antimicrobial agents for bacteria and fungi in planktonic form or biofilm. Their properties depend on: type of NPs, size, shape and capping agent type [78,82,86,98,99].

## 4. Safety Concerns

NPs have beneficial properties discussed in the previous sections of this article. But as any potential treatment, they will have side effects if used in clinical practice. Unfortunately, data on cytotoxicity of NPs against healthy bone cells is insufficient; only a few papers examined this aspect. Albers et al. have reported that 50 nm AgNPs can decrease the viability and proliferation rate of primary osteoblast and primary osteoclast [101]. Another study has shown that 15-nm AgNPs can trigger hFOB1.19 (human foetal osteoblast) apoptosis and necrosis via increased production of nitric oxide [102]. Also, AuNPs can influence the bone cell viability. AuNPs in the shape of rods and stars decreased the viability of hFBO1.19 cells, whereas the spherical-shaped ones did not [20]. Also, TiO_2_NPs may be harmful to the bone cells. TiO_2_NPs (10–15 nm) were internalised by hFOB1.19 cells and decreased their viability in a concentration-dependent manner by triggering oxidative stress [103]. We are unable to find any other data about in vitro cytotoxicity of NPs to healthy bone cells. Unfortunately, there was only one animal study regarding that matter. In in vivo (Wistar rats) assay, neither 20 nm AgNPs nor 21 nm TiO_2_NPs were toxic to red and white cells in the bone marrow [104]. Unfortunately, reticulocytes and leucocytes in the bone marrow responded negatively to AgNPs and TiO_2_NPs [104].

Although the available data are scarce, it is clear that NPs can be harmful. However, we should keep in mind that it is true for any other drug as well. Many commonly used antimicrobial agents (polymyxin B, amphotericin B, colistin M, cefazolin, ciprofloxacin, tetracycline, rifampicin, clindamycin, azithromycin, chloramphenicol, linezolid) can affect cell viability and/or proliferation [105,106,107]. Moreover, commonly used chemotherapeutics have numerous side effects; for example gemcitabine causes myelosuppression, hearing loss and liver failure, cytarabine damages the brain, heart and gastrointestinal tract and is also myelotoxic, and doxorubicin destroys bone marrow and causes nausea [108,109,110]. 

## 5. Clinical Usage

More than 51 products with nanotechnology developments are FDA approved [111]. Several products with hydroxyapatite or calcium phosphate in nanocrystal form were approved as bone substitutes (Vitoss^®^, Ostim^®^, OsSarura^®^, NanoOss^®^, EquivaBone^®^) [111]. Regarding matters discussed in this review the usage of carbon NPs and supramagnetic iron oxide NPs in lymph node biopsy [112,113] or medical imaging [111] is especially interesting. In all mentioned studies there were no information concerning side effects after application of NPs. 

## 6. Conclusions

Despite the recent advancements in orthopaedics bone cancers and implant-related infections are still unsolved problems. However, in the future, NPs may be applied as therapeutic agents. Because of their unique properties, both organic and inorganic NPs could potentially be used. In cancer therapy, NPs can be (I) directly cytotoxic to cancer cells, (II) drug delivery platforms or (III) hyperthermic agents. Moreover, NPs can be more effective than the drugs currently used in the clinic. As an adjuvant to the implant, NPs can (I) increase their biocompatibility by promoting osteogenesis and (II) be antimicrobial agents. Unfortunately, NPs can be also harmful to healthy cells. Several factors influence the biological properties of NPs (I) type of NPs, (II) concentration, (III) size, (IV) shape, (V) pH of environment, (VI) capping agents, (VII) functionalisation.

In future research, there is a need for a better understanding of the mechanisms of NPs biological properties, especially the antimicrobial ones. While focusing on the positive aspect of NPs in bioscience, we should also peruse nanotoxicological studies—the better we understand the NPs harmful effect the better we can avoid the side effects. Detailed knowledge about interaction between NPs and living cells in terms of cytotoxicity, anticancer and antimicrobial properties will allow designing nanoparticles-based drugs and biomaterials with highly favourable pharmacological/toxicological profile. Indisputably, NPs are a powerful tool, however there is still a lot to be done before we acknowledge that they can be used without any unknown risks. 

## Figures and Tables

**Figure 1 nanomaterials-10-00658-f001:**
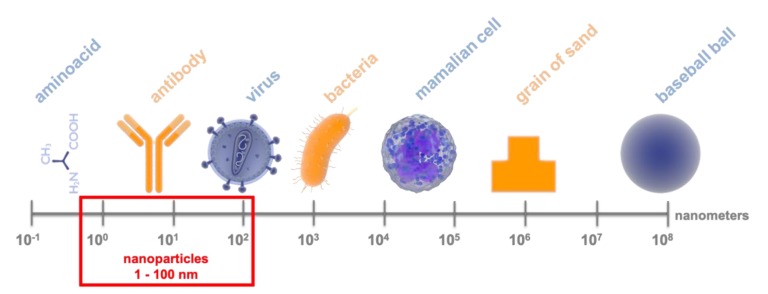
Comparison of nanoparticle size to other objects; presented on a logarithmic scale.

**Figure 2 nanomaterials-10-00658-f002:**
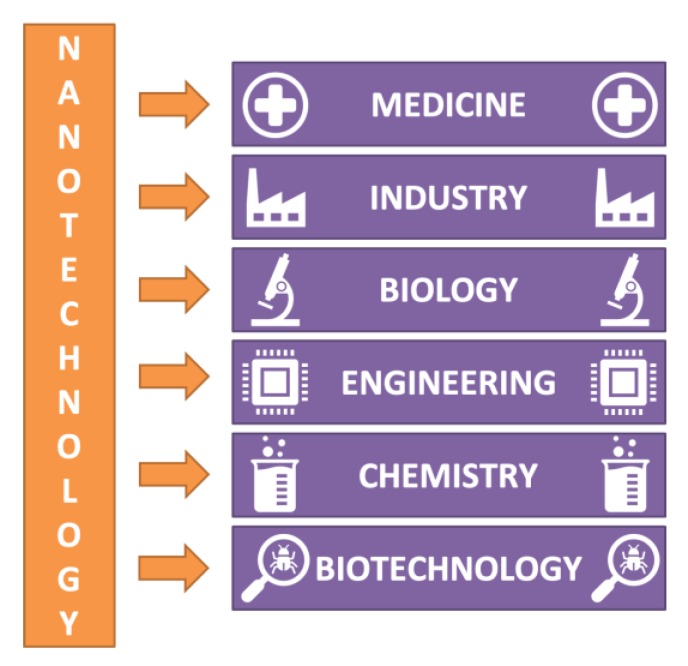
Applications of nanotechnology.

**Figure 3 nanomaterials-10-00658-f003:**
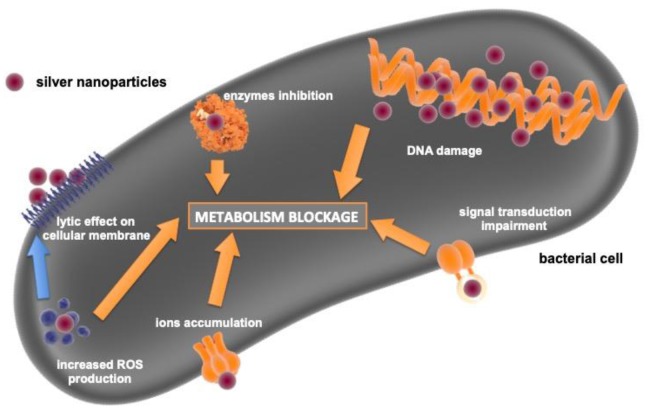
Schematic summary of AgNPs antibacterial activity mechanism.

**Table 1 nanomaterials-10-00658-t001:** Summary of nanoparticles (NPs) effects in in vitro model of osteosarcoma.

Nanoparticles Type	Osteosarcoma Cell Line	Effect	Additional Comment	Reference
Gold NPs 24.3 nm capped with advance glycation products	Saos-2	Cytotoxicity		[19]
Gold NPs rods Gold NPs stars Gold NPs spheres	143B MG63	Cytotoxicity Apoptosis induction	Cytotoxicity was shape-dependent	[20]
Citrate silver NPs 5 nm and 35 nm	U2OS Saos-2	Cytotoxicity Proliferation inhibition Mitochondrial stress and apoptosis induction	Cytotoxicity was size-dependent NPs were more effective than cisplatin	[23]
Copper NPs 10 nm	Saos-2	Cytotoxicity		[24]
Titanium oxide NPs 3,8 nm	U2OS	Cytotoxicity Increased ROS production Depletion of GSH		[25]
Titanium oxide NPs 15 nm	UMR-106	Cytotoxicity NPs were present in phagocytic vesicle within the cells		[26]
Aluminium oxide NPs 50 nm	UMR-106	Cytotoxicity NPs were present in phagocytic vesicle within the cells		[26]
Dextran coated cerium oxide NPs 3–4 nm	MG63	Cytotoxicity Increased ROS production	Cytotoxicity was pH-dependent Cells were more susceptible to NPs in an acidic environment	[27]
Zinc oxide NPs 22 nm	MG63	Cytotoxicity Increased ROS production Apoptosis induction		[28]
Cerium oxide NPs 26 nm	MG63	Cytotoxicity Increased ROS production Apoptosis induction		[28]
Fucoidan NPs 100 nm	C3H	Cytotoxicity Apoptosis induction	Fucoidan in NPs were more effective than fucoidan itself	[29]
Hydroxyapatite NPs 40 nm	MG63	Selective cytotoxicity only to cancer cells Ultrastructure changes	HA-NPs were cytotoxic to osteosarcoma cells and stimulated the growth of healthy osteoblast	[30]

**Table 2 nanomaterials-10-00658-t002:** Summary of NPs effects in in vitro model of fibrosarcoma.

Nanoparticles Type	Fibrosarcoma Cell Line	Effect	Additional Comment	Reference
Gold NPs 127 nm	HT-1080	Anti-metastatic effect	NPs did not affect cells viability AuNPs interfered actin-polymerisation pathway AuNPs inhabited cells migration	[34]
Silver NPs 6 nm	WEHI164	Cytotoxicity	IC_50_ of AgNPs was 2.6 μg/mL	[35]
Iron (II, III) oxide NPs 10 nm	HT-1080	Cytotoxicity	NPs had magnetic properties NPs may be used as drug delivery platform	[36]
Iron (II, III) oxide NPs 10 nm, 100 nm	HT-1080	Cytotoxicity Genotoxicity	NPs were coated with: -OH, -NH_2_, -TEOS, -AMPTS or TEOS/AMPTS functional groups Cytotoxicity and genotoxicity were function group – dependent AMPTS coated NPs were the most cytotoxic Positively charged NPs were more genotoxic than negatively charged	[37]
Cerium oxide NPs 25 nm	HT-1080	Non-cytotoxic		[38]
Cerium oxide NPs 30 nm	WEHI164	Cytotoxicity	Cancer cells were more susceptible to NPs than non-transformed ones NPs triggered oxidative stress NPs caused apoptosis	[39]
Chromium oxide NPs	L929	Cytotoxicity	NPs triggered oxidative stress NPs caused apoptosis	[40]

**Table 3 nanomaterials-10-00658-t003:** Summary of NPs properties as drug delivery treatment.

Nanoparticle Type	Cell Line	Drug	Comment	Reference
PGLA NPs	U2OS	Curcumin	NPs triggered mitochondria-dependent apoptosis	[16]
Streamline-dextran NPs	KHOS U2OS Drug-resistant osteosarcoma cells	Doxorubicin	The drug was more accumulated in drug-resistant cell lines Antiproliferative effect and apoptosis induction DOX in NPs were more effective than free drug	[17]
PEG NPs with stem-cell aptamer	Saos-2 U2OS MG63	Sialomycin	NPs were more effective against cancer cell line than non-cancerous cell	[18]
PEGylated PLGA NPs	MG63 Saos-2	Paclitaxel Etoposide	NPs were more effective than PTX and ETP in combination Apoptosis induction G2/M arrest	[32]
Dextran-g-PEI NPs	MG63 Saos-2	Adriamycin Plasmid DNA	Anticancer activity NPs were almost as good as typically used transfection reagent	[41]
Glutathione coated gold NPs	143B	Doxorubicin Gemcitabine Cytarabine	Cancer cell lines were more susceptible to NPs than non-transformed ones NPs conjugated with chemotherapeutic may be more effective than chemotherapeutic alone	[44]
Liposomal NPs	KHOS	Curcumin	Liposomal NPs with curcumin triggers apoptotic death whereas curcumin alone induces autophagy	[43]

**Table 4 nanomaterials-10-00658-t004:** Summary of antimicrobial activity of nanoparticles.

Nanoparticles Type	Microorganism	Comment	Reference
Silver NPs 75 nm	*Escherichia coli*	NPs had antibacterial activity.	[52]
Silver NPs 7 nm, 29 nm and 89 nm	*Escherichia coli* *Staphylococcus aureus*	MIC values were size-dependent. Bigger nanoparticles were less effective than smaller ones	[78]
Silver NPs 10–15 nm	*Escherichia coli* *Staphylococcus aureus* *Ampicillin resistant Escherichia coli* *Multi drug resistant Salmonella typhi*	Gram-negative bacteria are more susceptible to NPs NPs were effective against drug-resistant bacteria NPs inhibited signal transduction	[79]
Silver NPs Starch stabilised 20–40 nm *	*Staphylococcus aureus* *Pseudomonas aeruginosa* *Shigella flexneri* *Salmonella typhi* *Mycobacterium smegmatis*	NPs had antibacterial activity.	[80]
Lipolic acid- silver NPs 9.5 nm PEG- silver NPs 9.8 nm Tannic acid – silver NPs 10 nm Silver NPs 11.2 nm	17 different gram-negative strains 9 different gram-positive strains	Antimicrobial activity was capping agent dependent Gram-positive bacteria were more susceptible to NPs NPs had antibiofilm activity	[82]
Silver NPs 13,5 nm	*Escherichia coli* *Staphylococcus aureus* Yeast	NPs had antibacterial and antifungal activity	[84]
Silver NPs 25 nm	*Candida albicans* *Candida parapsilosis* *Candida tropicalis*	NPs stabilised with surfactants or polymers had higher antifungal activity The antifungal effect was present in non-cytotoxic concentrations	[86]
Silver NPs 3 nm	*Candida albicans* *Candida tropicalis* *Candida parapsilosis* *Candida krusei* *Candida glabrata* *Trichophyton mentagrophytes*	NPs were more effective than amphotericin B and fluconazole	[87]
Gold NPs (No size info)	*Escherichia coli*	NPs impacted expression of 359 genes NPs inhibited ATP synthesis and dissipated membrane potential NPs increased ROS production	[90]
Gold NPs 11–22 nm	*Listeria monocytogenes* *Bacillus cereus* *Staphylococcus aureus* *Escherichia coli* *Pseudomonas aeruginosa* *Salmonella typhimurium* *Candida albicans* *Aspergillus niger* *Aspergillus flavus*	NPs were effective against Gram-positive and Gram-negative bacteria NPs were more effective than ciprofloxacin against bacteria	[91]
Gold NPs 18.32 nm	*Staphylococcus aureus* *Pseudomonas aeruginosa*	NPs had antibacterial activity.	[92]
Copper NPs 62.5 nm	*Escherichia coli*	NPs caused dissipation of cell membrane, generation of ROS, lipid peroxidation, protein and DNA degradation in bacterial cells	[95]
Copper NPs (No size info)	*Micrococcus luteus* *Staphylococcus aureus* *Klebsiella pneumoniae* *Pseudomonas aerugionsa* *Aspergillus flavus* *Aspergillus niger* *Candida albicans*	NPs had antibacterial and antifungal activity	[96]
Zinc oxide NPs 200 nm	*Escherichia coli* *Listeria monocytogenes*	NPs had antibacterial activity.	[97]
Zinc oxide NPs 10 nm, 100 nm, 1 μm	*Candida albicans*	NPs antifungal activity was size-dependent NPs antifungal action is ROS mediated	[98]
Copper oxide NPs Titanium oxide NPs Zinc oxide NPs Aluminium oxide NPs Silicon oxide NPs Iron oxide NPs Cerium oxide NPs 25–50 nm	*Escherichia coli*	NPs antibacterial properties were material dependent (CuONPs > TiO_2_NPS > ZnONPs > Al_2_O_3_NPs > SiO_2_NPs > Fe_2_O_3_NPs > CeO_2_NPs) NPs antimicrobial activity was correlated with increased ROS production	[99]
Copper NPs 12 nm	*Escherichia coli*	NPs had antibacterial activity.	[100]

* No detailed size information.

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
