# Peer review of "Modified Nanoparticles as Potential Agents in Bone Diseases: Cancer and Implant-Related Complications"

_nanomaterials, 2020, doi:10.3390/nano10040658_

Round 1

Reviewer 1 Report

The subject is appealing and the Review is interesting and well written. Nevertheless, more care should be addressed to the Conclusions, a diffusely  read part of the manuscript, in particular in the Reviews. Conclusions should display a more conversational style, with no references -they are already reported along  the text- and, possibly, a general general comment.

Author Response

We would like to thank the Reviewer for their thoughtful comments and efforts towards improving our manuscript. Please find below point-to-point responses to the Reviewers’ critique.

Reviewer’s comment: Nevertheless, more care should be addressed to the Conclusions, a diffusely  read part of the manuscript, in particular in the Reviews. Conclusions should display a more conversational style, with no references -they are already reported along the text- and, possibly, a general general comment.

Authors’ response: As suggested we changed the conclusion sections.

“Although recent advancements in orthopaedics bone cancers and implant-related infections are still unsolved problems. However, in the future, NPs may be used as therapeutic agents. Due to their unique properties, both organic and inorganic NPs could potentially be used. In cancer therapy, NPs can be (I) directly cytotoxic to cancer cells, (II) drug delivery platforms or (III) hyperthermic agents. Moreover, NPs can be more effective than the drugs currently used in the clinic. As an adjuvant to the implant NPs can (I) increase their biocompatibility by promoting osteogenesis and (II) be antimicrobial agents. Unfortunately, NPs can be also harmful to healthy cells. Several factors influence the biological properties of NPs (I) type of NPs, (II) concentration, (III) size, (IV) shape, (V) pH of environment, (VI) capping agents, (VII) functionalization.

In future research, there is a need for a better understanding of the mechanisms of NPs biological properties, especially the antimicrobial ones. While focusing on the positive aspect of NPs in bioscience, we should also peruse nanotoxicological studies - the better we understand the NPs harmful effect the better we can avoid the side effects. Detailed knowledge about interaction between NPs and living cells in terms of cytotoxicity, anticancer and antimicrobial properties will allow designing nanoparticles-based drugs and biomaterials with highly favourable pharmacological/toxicological profile. Indisputably NPs are a powerful tool however there is still a lot to be done before we acknowledge that they can be used without any unknown risks.” (changes are underlined)

Reviewer 2 Report

The manuscript is understandable and helpful.   

a few spelling errors ae found.

Author Response

Reviewer’s comment: a few spelling errors ae found.

Authors’ response:

We would like to thank the Reviewer for their thoughtful comments and efforts towards improving our manuscript. 

We have carefully checked and corrected grammatical mistakes and typos in the manuscript. We changed spelling in lines: 26, 127, 183, 200, table 3, 290, 364, table 4 (indicated in track changes).

Reviewer 3 Report

Comments to the Authors:

In this manuscript, the authors reviewed nano-materials for applying in bone diseases and the molecular roles of these nano-materials. These studying may provide a possible application for osteosarcoma therapy and bone diseases. Some writings need to be further clarified. Overall, this manuscript needs a minor revision before publication. Detailed comments and suggestions are listed below.

Comments:

  1. The title of this manuscript should be modified.
  2. Generally, nanotechnology has been developed for a long time. Figure 1 may be replaced by the applications of nanotechnology.
  3. All the scale in nanoparticle size should be uniform.
  4. All the cell lines mentioned in this manuscript should be uniform (e.g., MG63 or MG-63; U2OS or U-2 OS).
  5. Page 3, Line 88: Are A-431 cells osteosarcoma. Please clarify.
  6. Page 8, Line 168, and page 10, Line 230: Please confirm the cited authors.

Author Response

We would like to thank the Reviewer for his thoughtful comments and efforts towards improving our manuscript. Please find below point-to-point responses to the critique.

Reviewer’s comment: The title of this manuscript should be modified.

Authors’ response: We shortened the title to “Modified nanoparticles as potential agents in bone diseases: cancer and implant-related complications”.

Reviewer’s comment: Generally, nanotechnology has been developed for a long time. Figure 1 may be replaced by the applications of nanotechnology.

Authors’ response: We are thankful for this comment. Instead of changing figure 1 we created figure 2 with summary of applications of nanotechnology.

Reviewer’s comment: All the scale in nanoparticle size should be uniform.

Authors’ response: We modified figure 1 according to reviewer’s suggestion.

Reviewer’s comment: All the cell lines mentioned in this manuscript should be uniform (e.g., MG63 or MG-63; U2OS or U-2 OS).

Authors’ response: We unified cell lines names (MG63 and U2OS). Changes had been made in lines 88, 123, 180 (indicated in track changes).

Reviewer’s comment: Page 3, Line 88: Are A-431 cells osteosarcoma. Please clarify.

Authors’ response: The information that A-431 is osteosarcoma cell line is taken from the cited paper (Mohanta, Y.K.; Panda, S.K.; Jayabalan, R.; Sharma, N.; Bastia, A.K.; Mohanta, T.K. Antimicrobial, Antioxidant and Cytotoxic Activity of Silver Nanoparticles Synthesized by Leaf Extract of Erythrina suberosa (Roxb.). Front. Mol. Biosci. 2017, 4, 14.). 

Reviewer’s comment: Page 8, Line 168, and page 10, Line 230: Please confirm the cited authors.

Authors’ response: In line 168 the author’s name is spelled correctly, unfortunately we made typo in line 230 which is now corrected (indicated in track changes). We are extremally sorry for that error.

Prof. Iwona Inkielewicz-Stępniak PhD